# cDNA Characterization and Expression of Selenium-Dependent *CqGPx3* Isoforms in the Crayfish *Cherax quadricarinatus* under High Temperature and Hypoxia

**DOI:** 10.3390/genes13020179

**Published:** 2022-01-20

**Authors:** Laura E. Hernández-Aguirre, Yazmin I. Fuentes-Sidas, Lizandro R. Rivera-Rangel, Néstor Gutiérrez-Méndez, Gloria Yepiz-Plascencia, David Chávez-Flores, Francisco J. Zavala-Díaz de la Serna, María del R. Peralta-Pérez, Antonio García-Triana

**Affiliations:** 1Molecular Biology Laboratory, Chemical Sciences Faculty, Circuit # 1 New Universitarium Campus, Autonomous University of Chihuahua (UACH), Chihuahua 31125, Chihuahua, Mexico; aguirre.laura@live.com (L.E.H.-A.); yasi_my86@hotmail.com (Y.I.F.-S.); lizandro_rrr20@hotmail.com (L.R.R.-R.); ngutierrez@uach.mx (N.G.-M.); dchavezf@uach.mx (D.C.-F.); fzavala@uach.mx (F.J.Z.-D.d.l.S.); mperalta@uach.mx (M.d.R.P.-P.); 2Research Center in Food & Development (CIAD), Gustavo Enrique Astiazarán Rosas Road, No 46, La Victoria Suburb, Hermosillo 83304, Sonora, Mexico; gyepiz@ciad.mx

**Keywords:** *Cherax quadricarinatus*, glutathione peroxidase, *GPx3*, gene expression, temperature, hypoxia, Pro-rich

## Abstract

Glutathione peroxidase 3 (GPx3) is the only extracellular selenoprotein (Sel) that enzymatically reduces H_2_O_2_ to H_2_O and O_2_. Two GPx3 (*CqGPx3*) cDNAs were characterized from crayfish *Cherax quadricarinatus*. The nerve cord *CqGPx3a* isoform encodes for a preprotein containing an N-terminal signal peptide of 32 amino acid residues, with the mature Sel region of 192 residues and a dispensable phosphorylation domain of 36 residues. In contrast, the pereiopods *CqGPx3b* codes for a precursor protein with 19 residues in the N-terminal signal peptide, then the mature 184 amino acid residues protein and finally a Pro-rich peptide of 42 residues. *CqGPx3* are expressed in cerebral ganglia, pereiopods and nerve cord. *CqGPx3a* is expressed mainly in cerebral ganglia, antennulae and nerve cord, while *CqGPx3b* was detected mainly in pereiopods. *CqGPx3a* expression increases with high temperature and hypoxia; meanwhile, *CqGPx3b* is not affected. We report the presence and differential expression of *GPx3* isoforms in crustacean tissues in normal conditions and under stress for high temperature and hypoxia. The two isoforms are tissue specific and condition specific, which could indicate an important role of *CqGPx3a* in the central nervous system and *CqGPx3b* in exposed tissues, both involved in different responses to environmental stressors.

## 1. Introduction

Regulation of reactive oxygen species (ROS) is critical for all known organisms. ROS are important regulators of metabolism and exert essential physiological functions. Hydrogen peroxide is a ROS that is produced from a superoxide ion, and its modulation is given to the enzymatic level from peroxidases that hydrolyze H_2_O_2_ to H_2_O and O_2_. Although most peroxidases have heme in their active site, some others contain selenocysteine (Sec). Selenoproteins (Sels) are a family of proteins that contain Sec [1], where SECIS—a cis-acting selenocysteine insertion sequence in their mRNA—is essential for Sec incorporation into the proteins [2]. Glutathione peroxidases (GPxs) are selenoproteins that catalyze the reduction of H_2_O_2_ to water, usually using glutathione (GSH) as a reducing agent [3]. Sec in the active site of glutathione peroxidase is encoded by the termination codon TGA [4], and is the cause of increased affinity for hydroperoxide and reducibility for GSH.

GPxs are important components of the REDOX system, but until now only a few genes have been reported in crustaceans [5]. The GPx from penaeid shrimp *Metapenaeus ensis* (*MeGPx*) has high similarity with human glutathione peroxidase 3 (GPx3) [6]. Selenium-dependent glutathione peroxidases (*Se-GPx*) were also identified from the shrimp *Fenneropenaeus chinensis* and *Litopenaeus vannamei*, both similar to GPx1 from some species [7,8], while the *GPx* cDNA from *Procambarus clarkii* (*PcGPx*) shares high similarity with vertebrate GPx1 and GPx2 [9]. In addition, of the two known *Penaeus monodon* GPxs (GPx1 and GPx7), only GPx1 was reported as a selenoprotein [10].

In contrast to the few GPxs identified in crustaceans, at least eight glutathione peroxidases (GPx1–GPx8) are known in mammals, and GPx3 is the only known Sec-dependent extracellular and membrane-associated GPx [11]. *GPx3* expression reduces extracellular H_2_O_2_ concentration in human muscle cells regulating H_2_O_2_ levels and its function as a second messenger, and the inhibition of *GPx3* is associated with unbalance in the redox system, insulin resistance and diabetes mellitus [12]. Particularly, GPx3 in the nervous and muscular systems is associated with pathogen resistance, survival and inflammatory response. For instance, *GPx3* and thioredoxin-like 1 (*TXNL1*) expression are induced by the transcription factor NeuroD6 in the spinal cord, producing the ROS depletion involved in the attenuation of inflammation [13]. Furthermore, knockdown of *GPx3* in human skeletal muscle precursor cells using siRNA resulted in a higher production of reactive oxygen species [14], while the expression of *GPx3* was threefold higher in fat cells compared to non-fat cells of continuously stressed adipose tissue from morbidly obese women, indicating an inflammatory response [15].

Hypoxia alters *GPx3* expression in a tissue-specific manner in mammals. Human bone marrow mesenchymal stem cells cultured in hypoxic conditions showed an increase in cell fitness, evidenced by an improvement in clonogenicity and improved differentiation potential towards adipocyte and chondrocyte lineages. Under these circumstances, *GPx3* expression rose 4.5-fold [16]. In the butterfly-shaped interscapular brown adipose tissue (iBAT) of myoglobin knockout (MBko) mice, *GPx3* expression decreased significantly compared to the wild type, indicating anoxia-related diminishment of H_2_O_2_ and a ATP synthesis decrease [17]. Variants or isoforms of *GPx3* have been reported in differential expression patterns. In kidney embryonic cells HEK-293, *GPx3* was down-regulated in thioredoxin reductase 1 variant 1 (TXNRD1_v1) overexpressing cells; interestingly, no change in *GPx3* expression was identified in thioredoxin reductase 1 variant 2 (TXNRD1_v2) overexpressing cells [18]. Furthermore, *GPx3* exhibits a high expression level in aquatic animals in response to biotic and abiotic stressors such as handling and high temperature [19], bacterial infection [20] and toxin detoxification [21,22], probably playing a major role in the balance of oxidative stress.

On the other hand, since crayfish lack the adaptive immunity of vertebrates, they depend totally on their innate immune system to fight infections. Antimicrobial peptides (AMPs) are one of their lines of defense, as in other crustaceans, usually inhibiting protein, RNA or DNA synthesis to kill microorganisms with expression cell or tissue dependence [23,24]. Pro-rich is an AMP that kills bacteria by binding to ribosome and chaperone proteins [25]. Astacidin is an AMP of 20 amino acids Pro-rich peptide identified in *P. clarkii* with this function, capable of antimicrobial activity against *Staphylococcus aureus* and *Vibrio anguillarum* [26]. It is known that temperature can affect the REDOX response in this crustacean.

A significant increase in total glutathione and lipid peroxidation occurred when these animals were exposed to 29 and 33 °C [27], but no antioxidant genes expression has been studied under stressful temperatures. Additionally, in *Macrobrachium nipponense*, transcriptomic analysis revealed that under hypoxia, the expression of oxidoreductase genes was higher [28]. Additionally, anoxia increased the NADPH concentration by the activation via phosphorylation of glucose-6-phosphate dehydrogenase in the crayfish *Orconectes virilis* [29]. No gene expression studies are available in *C. quadricarinatus* under hypoxia. However, in normal conditions, transcriptomic analysis in gills reported the presence of antioxidant coding genes [30], and although reduced GSH was lower in starved animals [31] no glutathione peroxidase gene has been specifically reported in these animals.

The objective of the present work was to analyze two GPx isoforms in *C**. quadricarinatus* nervous tissues in normal and stress conditions. We report the complete cDNA and deduced proteins sequences of *GPx3a* (*CqGPx3a*) and *GPx3b* (*CqGPx3b*), their expression in nerve cord (NC), cerebral ganglia (CG), antennule (Ante), ocular peduncle (OP), pereiopods (Pe), pleopods (Pl) and antennae (Ant). Finally, tissue-specific and stress-specific expressions of these isoforms under temperature and hypoxia stress were detected for the *GPx3* isoforms.

## 2. Materials and Methods

### 2.1. Animals and Bioassays for Temperature and Hypoxia

Sixty *C. quadricarinatus* male juveniles (21 ± 1.5 g) were donated by Biohelis, CIBNOR, Baja California Sur, Mexico. Crayfishes were acclimated for two weeks in 50 L freshwater plastic tanks at 23.3 ± 1.26 °C, with constant aeration (7.86 ± 0.73 mg/L dissolved oxygen), 12 h light: 12 h dark photoperiod conditions, culture density of 14 m^−2^, and fed ad libitum daily with commercial feed (Golden Bites, Biomaa, México). Only healthy inter-molt juveniles were chosen for all the experimental groups. Before the experiments, crayfishes were separated in 18 L glass aquaria. One-third of the water volume was changed twice daily to prevent ammonia accumulation; uneaten food particles and feces were removed daily. Optimum conditions experiment used to determine the expression of *CqGPx3* (*n* = 8, *N* = 56), *CqGPx3a*, and *CqGPx3b* isoforms (*n* = 8, *N* = 96) in a tissue-specific manner in the nervous system and muscle were kept at 23.0 ± 1.01 °C, with constant aeration (7.80 ± 0.56 mg/L dissolved oxygen), photoperiod conditions of 12 h light: 12 h dark, culture density of 14 organisms m^−2^, and fed ad libitum daily with commercial feed (Golden Bites, Biomaa, México). To analyze the expression of *CqGPx3a* and *CqGPx3b* isoforms (*n* = 4, *N* = 15) in a tissue-specific manner under stressful conditions, a control group and three stressful conditions groups were subjected to hypoxic stress (3 ± 1.5 mg/L DO), moderate temperature stress (27.5 ± 1 °C) and severe temperature stress (LD_50_) (30 ± 2.1 °C). A photoperiod of 12 h light: 12 h dark, culture density of 14 organisms m^−2^, and feeding ad libitum daily with commercial feed (Golden Bites, Biomaa, Mexico) conditions were kept in all experiments. The control group was held at 23.2 ± 1.12 °C and constant aeration (7.83 ± 0.62 mg/L dissolved oxygen). Hypoxia was induced by modulating aeration in the aquarium and was monitored with a dissolved oxygen meter (YSI 55, Yellow Spring, OH, USA). The temperature was adjusted with a submersible thermostatic heater, and the variables were measured three times a day, with adjustments made if necessary. After two weeks of stress challenge, tissues of interest were dissected and processed.

### 2.2. RNA Isolation, Amplification and Cloning of Partial GPx cDNA Fragments

Nerve cord (NC), cerebral ganglia (CG), antennule (Ante), ocular peduncle (OP), pereiopods (Pe), pleopods (Pl) and antennae (Ant) were dissected from crayfishes and kept at −80 °C until used. Total RNA was extracted using TRI^®^ (Sigma-Aldrich, San Luis, MO, USA) according to the instructions of the manufacturer. RNA integrity was confirmed by 1% agarose gel electrophoresis. Reverse transcription (RT) was performed using Quantitect Reverse transcription (Qiagen, Hilden, Germany). Next, 1 µg of total RNA was reverse transcribed with oligo dT (20 mer). Degenerate primers were designed based on conserved amino acid identified from the sequence alignments of related GPxs. Internal cDNA partial amplification of NC, CG, Ante, OP, Pe, Pl and Ant were obtained using different combinations of GPxF4, GPxF5, GPxF6, GPxR3 and GPxR4 primers (Table 1). The 25 µL reactions containing 12.5 µL AmpliTaq Gold^®^360 Master Mix, 0.5 µL Fw primer (10 µM), 0.5 µL Rv primer (10 µM), 10.0 µL Milli Q water and 1 µL of the respective cDNA under the following conditions: 94 °C for 5 min, followed by 94 °C for 1 min, 47 °C for 1 min and 72 °C for 1 min per cycle for 38 cycles and finally, 72 °C for 7 min in a DNA Thermal Cycler (Bio-Rad T100). PCR products were cloned in the pGEM-T Easy Vector System I (Promega, Madison, WI, USA) and transformed in DH5α Escherichia coli competent cells (Invitrogen, Carlsbad, CA, USA). Recombinant plasmids were sequenced in both DNA directions by the dideoxy chain-termination method (Macrogen Co., Seoul, Korea). cDNA sequence and deduced proteins were compared to nucleotide and protein GenBank databases using the BLAST algorithm [32] at the National Center for Biotechnology Information, Bethesda, MD.

### 2.3. CqGPx3a and CqGPx3b Rapid Amplification of cDNA Ends (RACE) for Nerve Cord and Pereiopods

Fragments containing 5′ and 3′ cDNA ends were obtained using SMARTer RACE (Invitrogen, Carlsbad, CA, USA). Approximately 1 µg of total RNA from NC or Pe was used for cDNA synthesis following the manufacturer’s recommendations. Based on a known partial sequence of *CqGPx3a* or *CqGPx3b* isoforms, the specific primers GPxF7, GPxF19, GPxR5 and GPxR12 were designed as shown in Table 1. For the 5′ and 3′ ends, amplification of *CqGPx3a* R5 and F7 were used, and R12 and F19 for the 5′ and 3′ ends amplification of *CqGPx3b*, respectively, following the manufacturer’s recommendations (63 °C of alignment temperature and 3 min to elongation). RACE-PCR products were cloned and sequenced for both DNA chains as mentioned above. cDNA sequences and the deduced proteins were compared to nucleotide and protein databases using BLAST. Complete cDNA sequences analyses were made with ExPASy bioinformatic tools available online: http://web.expasy.org/translate/ (accessed on 5 October 2021). SECIS secondary structures were deduced using SECISearch 2.19 algorithm available online: http://genome.unl.edu/SECISearch.html (accessed on 5 October 2021).

### 2.4. CqGPx3a and CqGPx3b mRNA and Deduced Amino Acid Sequence Analysis

Amino acids and glutathione peroxidase domain were deduced using the ExPASy translate tool and SMART sequence identifier, respectively, available online: http://smart.embl-heidelberg.de (accessed on 5 October 2021). Mature CqGPx3 proteins 3D model based on human glutathione peroxidase 3 were obtained using RaptorX available online: http://raptorx.uchicago.edu/ (accessed on 5 October 2021). SMART sequence identifier was used to identify putative N-terminal signal peptide and extracellular NIDO domain. Prosite ExPASy was used to identify possible carboxyl-terminal phosphorylation sites available online: http://prosite.expasy.org/prosite.html (accessed on 5 October 2021). The SECISearch 2.19 program was used to predict UTR 3´ selenocysteine insertion sequence (SECIS) available online: http://genome.unl.edu/SECISearch.html (accessed on 5 October 2021). Alignments were performed with CLUSTAL X 2.0 [33] and the visualization and statistical report were performed with GeneDoc 2.7.0 [34]. The LG + G model (0.442 for the γ distribution) was selected by the model test of MEGA for the tree search. The confidence at each node was assessed by 500 bootstrap replicates and the phylogenetic tree was constructed using Maximum Likelihood with MEGA 6.0 software [35].

Pro-rich peptides tertiary hypothetical structure was predicted by Phyre2 and RaptorX available online: http://www.sbg.bio.ic.ac.uk/phyre2/ http://raptorx.uchicago.edu respectively (accessed on 5 October 2021). Physicochemical properties, Grand Average hydropathy value (GRAVY) and Boman Index were predicted using APD3 available online: https://aps.unmc.edu/ (accessed on 5 October 2021). ProtScale software was used to calculate Kyte–Doolittle and Hopp–Woods scales available online: https://web.expasy.org/protscale/ (accessed on 5 October 2021). Amino acids flexibility probability was calculated using FlexPred and structure homology flexibility was predicted in PredyFlexy available software online: http://flexpred.rit.albany.edu/ http://www.dsimb.inserm.fr/dsimb_tools/predyflexy/ (accessed on 5 October 2021). Peptide cutter and PROSPER software were used to identify serine endopeptidases excision sites, available online: http://web.expasy.org/peptide_cutter/ https://prosper.erc.monash.edu.au/ (accessed on 5 October 2021).

### 2.5. CqGPx3, CqGPx3a and CqGPx3b mRNA Isoforms Quantification

Total RNA was extracted from NC, CG, Ante, OP, Pe, Pl and Ant of juvenile male crayfishes (*n* = 8, *N* = 56) using TRI^®^ (Sigma-Aldrich). Next, 1 μg of total RNA was reverse transcribed using QuantiTect Reverse Transcription Kit (Qiagen, Hilden, Germany) with oligo dT (20 mer) according to the manufacturer’s instructions. *CqGPx3* transcripts were amplified using the primers GPxF2 and GPxR2 (309 pb amplicon) (*n* = 8, *N* = 56). *CqGPx3a* isoform transcripts were amplified with GPxF21 and GPxR15 primers (258 pb amplicon) and *CqGPx3b* isoform transcripts were amplified with GPxF19 and GPxR12 primers (169 pb amplicon) (*n* = 8, *N* = 96). cDNAs were used as a template and qPCR was performed on a CFX96 Real Time PCR (Bio-Rad, Hercules, CA, USA) in two separated PCR reactions for each individual crayfish and tissue in 20 μL final volume contained 10 μL of 2X iQ SYBR Green Supermix (Bio-Rad), 8.5 μL of H_2_O, 0.25 μL of each primer (10 μM) and 1 μL of cDNA (equivalent to 50 ng of total RNA). The ribosomal protein *L12* (GenBank accession no. AEL23104.1) cDNA was amplified side by side for comparisons using the forward and reverse primers L12F1 and L12R1 (amplicon 300 pb) under the same conditions. The RT-qPCR procedure consisted of 95 °C for 3 min, followed by 95 °C for 30 s and 64.1 °C for 35 s per cycle for 40 cycles, then a melting curve analysis was carried out by a slow increase (0.5 °C/5 s) from 65 to 95 °C, to examine primer-dimers presence and non-specific amplification. Standard curves for *CqGPx3*, *CqGPx3a*, *CqGPx3b* and *L12* were run to determine the efficiency of amplification using dilutions from 5 × 10^−4^ to 5 × 10^−9^ ng/μL of purified amplicons. For each measurement, expression levels (ng/μL) were normalized to *L12* and expressed as relative expression values (*CqGPx3/L12*, *CqGPx3a/L12*, *CqGPx3b/L12*).

### 2.6. Statistics

One-way analysis of variance (ANOVA) and Tukey and LSD test (α < 0.05) were applied to the data. The Minitab 17 statistical software was used.

## 3. Results

### 3.1. Two Isoforms of GPx3 Were Identified in Nervous System Tissues

Using GPxF2 and GPxR2 primers (Table 1), PCR fragments of *CqGPx3* cDNA were obtained from NC, CG, Ante, OP, Pe, Pl and Ant and sequenced. Two isoforms were identified and denominated as *CqGPx3a* and *CqGPx3b*. The *CqGPx3a* predicted protein partial sequence is identical in NC, CG and Ante, while the *CqGPx3b* deduced protein partial sequence is identical in Pe, Pl and Ant.

The cDNA complete sequence of *CqGPx3a* from NC is 1082 bp (Figure 1). It does not have a 5′ untranslated region (UTR) and has a coding sequence of 780 bp corresponding to 260 amino acids and 29.26 kDa, with a stop codon at position 781, and a 302 bp 3′ UTR. The characteristic Sec UGA insertion codon was identified at position 237. The 3′ UTR contains a consensus polyadenylation signal (AAUAAA) and a poly-A tail. In the 3′ UTR, a SECIS type 2 structure was predicted. A putative pro-peptide comprises the signal peptide identified by SMART (32 amino acids), the mature *CqGPx3a* protein of 192 amino acids and two asparagine N-linked glycosylation sites (amino acids 56 and 120) were identified using Prosite. The N-terminal signal peptide cleavage site is at 32 amino acid (GLG-KI) and the C-terminal cleavage site is at 224 amino acid (LY-EL). In the C-terminal domain, there are two different recognition phosphorylation sites identified in the PPEVP and RRIS sites by Prosite ExPASy (36 amino acids). Complete pro-peptide modeling is important to determine the non-catalytically viable characteristics of the protein. In this sense, the C-terminal 228 residues RaptorX analysis shows *CqGPx3a* with (100% of the sequence) were modeled with *p*-value: 9.83 × 10^−11^, uGDT(GDT): 146(64) and uSeqId(SeqId): 85(37). Adequate folding is presumed when serin proteases liberate the C-terminal in natural circumstances. For *CqGPx3a* without C-Terminus, 192 residues (100% of the sequence) were modeled with *p*-value: 6.47 × 10^−11^, uGDT(GDT): 145(76) and uSeqId(SeqId): 84(44). *CqGPx3a* proteins were modeled using human GPx3 (selenocysteine to glycine mutant) as template with 1.85 Armstrong resolution and 100% confidence. The newly found GPx was named *CqGPx3a* and was deposited in GenBank under accession no. KX685410.

The full-length *CqGPx3b* is 1025 nt. The cDNA nucleotide and deduced amino acid sequences are shown in Figure 2. It contains a 45 bp 5′ UTR, 735 bp coding sequence (245 amino acids) and 245 bp in the 3′ UTR. The initiation methionine codon (ATG) is found at nucleotide 46 and a stop codon is present at position 781. *CqGPx3b* contains a TAG codon (244 nt), corresponding to Sec residue at the 67th codon (U).

Analysis of both *CqGPx3* SECIS predicted conformation showed a typical type 2 structure with an additional helix upstream of the adenosine bulge (Figure 3). The putative pro-peptide comprises the signal peptide (19 amino acids), the mature *CqGPx3b* protein (184 amino acids) and the C-terminal domain (42 amino acids). SMART domains analyses showed an N-terminal signal peptide from amino acid 1 to 19 and a cleavage site (GLG-EI, probability of 0.992) in the SignalP 3.0 Server. Prosite identified asparagine N-linked glycosylation sites (amino acid positions 43 and 107). The C-terminal cleavage site is present at residue 203 (LK-SD). GPx-conserved motifs PCNQF, VNG and WNFEKFL are presented in the *CqGPx3b* sequence (Figure 2) [7]. *CqGPx3b* complete pro-peptide modeling is also important to determine the non-catalytically viable characteristics of the protein. RaptorX analysis shows *CqGPx3b* with C-Terminus; 226 residues (100% of the sequence) were modeled with *p*-value: 1.26 × 10^−10^, uGDT(GDT): 143(63) and uSeqId(SeqId): 83(37). Additionally, adequate folding is presumed when serin proteases liberates the C-terminal in natural circumstances. *CqGPx3b* without C-Terminus, 184 residues (100% of the sequence) were modeled with *p*-value: 6.60 × 10^−11^, uGDT(GDT): 136(74) and uSeqId(SeqId): 82(45). *CqGPx3b* proteins were modeled using human GPx3 (selenocysteine to glycine mutant) as template with 1.85 Armstrong resolution and 100% confidence. The newly found GPx gene was named *CqGPx3b* and was deposited in GenBank under accession no. KX685411. The calculated molecular mass of *CqGPx3b* was 27.06 kDa. *CqGPx3a* and *CqGPx3b* Selenocysteine Insertion Sequence (SECIS) have putative type 2 secondary structures (Figure 3).

Alignment of the *CqGPx3a* and *CqGPx3b* predicted proteins with related crustaceans and GPx3s of mammals and amphibians indicate that these enzymes conserved the signature sequences. *P. clarkii* and *M. ensis* GPx are not classified as GPx3 and are the only ones that do not present the characteristic N-terminal signal peptide. Additionally, a 40 amino acids C-terminal difference can clearly be recognized among CqGPxs and GPx3. *CqGPx3a* and *CqGPx3b* have 44 and 46% identity with *P. clarkii* GPx (G9JJU2.1), 31 and 34% with *M. ensis* GPx (ACB42237.1), 55 and 57% with *P. monodon* GPx (ALM09356.1) and 34 and 36% with human GPx3 (P22352.2), respectively (Figure 4).

The three-dimensional structure of *CqGPx3a* and *CqGPx3b* with and without the C-terminal domain analyzed with RaptorX indicates protein structure changes in the presence or absence of the C-terminal domain (Figure 5). In both cases, the catalytic site is not available while the proteins are in the pro-peptide form, indicating a possible lack of activity.

Phylogenetic analysis showed that *CqGPx3a* and *CqGPx3b* isoforms are clustered with MeGPx, PcGPx and *P. monodon* GPx3 (Figure 6).

### 3.2. CqGPx3b C-Terminal Domain Prediction of Pro-Rich Antimicrobial Function

In the C-terminal sequence of *CqGPx3b*, there is a 30-amino-acids fragment that has a proline rich (Pro-rich) region. In this sequence, 10 amino acids have transmembrane region homology. Phyre^2^ and RaptorX software modeled Pro-rich peptide as random coiled tertiary structure using a hydrolase loop as a template (NDB ID: NA1014) (Figure 7). The Pro-rich peptide is enriched with five proline residues (Pro-rich) and six valine residues (Val-rich). The probability of target multiplicity increases by the <1 (0.83 kcal/mol) Boman index result. The Pro-rich peptide has a 40% hydrophobic calculated region that forms an N-terminal hydrophobic core. The Liu–Deber value of transmembrane region homology is 0.59. The GRAVY Peptide solubility value is 0.1. N-terminal hydrophobic and C-terminal hydrophilic cores were determined using Kyte–Doolittle and Hopp–Woods scales in ProtScale software. Peptide cutter and PROSPER software were used to identify K (117), A (119) and Q (151) serine endopeptidases excision sites. Although three different C-terminal peptides may be released, the analysis showed that a 31-amino-acids length peptide has more possibilities to be the functional form.

### 3.3. Expression of CqGPx3, CqGPx3a and CqGPx3b Isoforms in a Tissue-Specific Manner in the Nervous System and Muscle

Previously to the dissection and processing of the tissues of interest, a mobility behavior reduction was identified in crayfishes subjected to hypoxia. No behavioral differences compared to the control group were identified in the 27.5 ± 1 °C temperature group, and the 30 ± 2.1 °C temperature group presented a mobility behavior reduction. The mean cDNA Ct value of the ribosomal protein *L12* for NC, CG, Ante, OP, Pe, Pl and Ant was 22.39 ± 2.03, therefore *L12* has a comparable expression in all the tissues studied. *CqGPx3* relative to *L12* RT-qPCR analysis from different tissues (NC, CG, Ante, OP, Pe, Pl and Ant) of *C. quadricarinatus* is presented proportionally to NC expression. Higher transcript levels were detected in CG (1.7-fold in respect to NC), followed by Pe and NC. NC expression is statistically different to CG, Ante and OP. Ant, Pl and Ante do not have significant differences in expression. A very low quantity in OP and different from all other tissues was detected (Figure 8).

The relative quantification analysis of *CqGPx3a* and *CqGPx3b* mRNA isoforms from the same tissue, excluding OP, indicates that both isoforms are expressed in all tissues with the exception of *CqGPx3a* in Pe and *CqGPx3b* in NC (Figure 9). The highest relative level of *CqGPx3a* was identified in CG followed by Ante and NC (no significant differences). The relative expression in Ant and Pl was 15 and 43-fold lower than CG, respectively. Conversely, *CqGPx3b* was expressed mostly in Pe, followed by Ant, Ante and CG (no significant differences) while the expression found in Pl was five-fold lower than Pe. *CqGPx3a* is preponderant over *CqGPx3b* in CG, NC and Ante; meanwhile, *CqGPx3b* is mainly found in Pe. No differences between *CqGPx3a* and *CqGPx3b* were identified in Pl and Ant ANOVA (*p* = 0.05).

### 3.4. CqGPx3a and CqGPx3b Expression in Stressful Conditons

Different expression patterns of *CqGPx3* isoforms were found under heat stress and oxygen deficiency (Figure 10). *CqGPx3a* expression increased in NC and Pe during temperature and hypoxia stress (*p* = 0.05) but did not have a significant effect on *CqGPx3b* expression in NC and Pe (*p* = 0.05). In NC, *CqGPx3a* expression increased 2.5-fold and 0.5-fold when exposed to moderate or severe temperatures and four-fold times when exposed to hypoxia; *CqGPx3b* expression was 900 to 4600-fold lower in NC than *CqGPx3a* under all stressful conditions (*p* = 0.05). *CqGPx3a* expression in Pe changes from undetectable in normal conditions to detectable with differences related to *CqGPx3a* in NC (*p* = 0.05). No change in Pe *CqGPx3b* expression was detected among control and temperature and hypoxia-stressed animals (*p* = 0.05). Interestingly, Pe *CqGPx3b* expression was significantly superior in the control group compared to *CqGPx3a* (*p* = 0.05). In stressful conditions, *CqGPx3a* is still the more abundant isoform expressed in NC.

## 4. Discussion

Glutathione peroxidases are important components of the antioxidant system in most cells. Until now in crustaceans, there are few reports of genes or cDNA sequences of these enzymes and to our knowledge, this is the first report of extracellular GPxs that are selenoproteins in crustaceans. In these two SeGPxs of *C. quadricarinatus*, the incorporation of SeCys in the TGA codon, instead of termination, is clearly indicated by the SECIS elements in the 3′ UTR. Two forms (1 and 2) of SECIS are known, with form 2 being more common than form 1 [36]. In *CqGPx3* isoforms, this SECIS appears to conform to the type 2 SECIS. Moreover, both *SeGPxs* are expressed in the nervous system, while the previously identified GPxs in crustaceans, such as MeGPx and PcGPx, are expressed in ovary, but not in the nervous system [6,9]. Similarly, no PcGPx expression was detected in brain, ganglia and muscle from the red swamp *P. clarkii* [9] and also, no MeGPx expression was detected in brain and ganglia of the shrimp *M. ensis* [6]. On the other hand, transcripts of a peroxidase selenoprotein M (*SelM*) were reported in *L. vannamei* Pl and muscle [37].

Some interesting features were found in the GPxs from *C. quadricarinatus*. The N-terminal signal peptides indicates that both, *CqGPx3a* and *CqGPx3b* are secreted proteins that contain two and one predicted N-glycosylation sites, respectively, and it is known that asparagine N-linked glycosylation sites are associated with secreted or membrane-bound proteins [38]. Additionally, the *CqGPx3a* 36 amino acids C-terminal is possibly a phosphorylation-regulated extension of the mature protein or may have a cleavable function-associated domain of the protein in the GPx, as have the PCNQF-, VNG- and WNFEKFL-conserved motifs (Figure 1) [7]. It is known that GPx1 phosphorylation induces stimulation of enzyme activity [39] and p47phox activation is induced by the phosphorylation of at least three C-terminal sites that relaxes the proteins’ constrained conformation [40].

N-terminal and C-terminal evidence, in addition to tissue-specific ovary but not nervous system expression [6,9], suggests that although MeGPx and PcGPx are related to CqGPx3s, they could have different metabolic functions. The characteristic presence of N-terminal signal peptide and cleavage in the predicted sequences supports the hypothesis that *CqGPx3a* and *CqGPx3b* are isoforms of extracellular GPx3. In humans, glutathione peroxidase 3 is the only extracellular GPx [11]. The *CqGPx3a* and *CqGPx3b* isoforms possibly have a similar function.

Structural pro-enzyme and enzyme differences are a strategy to regulate mature protein availability and activity. The finding of two isoforms of a single *GPx3* in the same species is interesting. A single-nucleotide change in GPx3 sequence can generate important availability consequences. For instance, serum levels of GPx3 are higher in subjects of a Mexican population with metabolic syndrome and cardiovascular risk associated with rs8177409 single-nucleotide polymorphism [41]. Thus, CqGPx3s may be involved in similar regulation processes, and the presence of the C-terminal domain could be associated with the regulation of its function and activity. Originally, MeGPx was clustered with vertebrate GPx3 and GPx5 [6] and PcGPx was clustered with vertebrate GPx1 and GPx2 rather than with GPx3 and GPx5.

The fragment enriched with prolines at the C-terminal end of *CqGPx3b* pro-peptide of *C. quadricarinatus* indicates its possible function as an antimicrobial peptide [23]. Unfolded structure infers about a high range of target molecules and it also propitiates the internalization to pathogens [42]. Proline abnormal content suggests an active antimicrobial peptide (AMP) function [24]. Additionally, valine enrichment can be associated with target cell selectivity and reduction in hemolysis and cytotoxicity against host cells [43]. AMP enriched with proline and valine (Pro/Val-rich) simultaneously has not been known. A Pro-rich APD3 software calculated -1 net charge indicates different action mechanisms or targets from preponderantly cationic, previously reported AMPs [44]. Boman index suggests that Pro-rich has a higher antimicrobial activity with fewer secondary effects [45]. An N-terminal hydrophobic core is needed to establish the primary contact with the target cells [46]. A Liu–Deber scale of 0.59 (threshold: 0.4) and its transmembrane region homology indicates that this fragment could act as a natural transmembrane mimetic that would lead to insertion into bacteria membrane [47]. GRAVY index value denoted Pro-rich peptide solubility in water and decreased cytotoxicity [48]. Amphipathicity has been described as the most valuable property for AMPs activity [49]. FlexPred and PredyFlexy software indicate that residues 18 to 30 have high possibilities of flexibility. Proline residues in the C-terminal end could propitiate transition to α helix structure when the peptide interacts with targets [50]. Pro-rich peptide cleavage and liberation in extracellular nervous system space by serine endopeptidases is highly probable in crayfishes due to their presence in *C. quadricarinatus* [51]. Pro-rich can potentially act over a wide range of microbial pathogens as a protein synthesis inhibitor, binding to the ribosome or the chaperone protein Dnak [25] without causing hemolysis or cytotoxicity in eukaryotic cells. Previously, a Pro-rich AMP presence in crustaceans was reported as astacidin, a 20 amino acids Pro-rich AMP that was found in *P. clarkii* [26]. Distinctive properties of proposed Pro-rich peptide as antimicrobial, suggest the presence of a new AMP group with high potential of applicability.

The abundance of the *GPxs* in the *C. quadricarinatus* nervous system infers an important role. It is known that GPxs have a protecting role in the nervous system. The increase in survival and protection has been proven in a neuroblastoma cell line infected with a lentivirus vector carrying the coding sequence for human GPx1 pLV-GPX1 [52]. Moreover, specifically for *GPx3*, the mRNA expression is increased in the aquatic animals *Hydra magnipapillata* and *Oryzias javanicus* in response to exposure to fluoxetine, a selective serotonin reuptake inhibitor, which possibly helps buffer oxidative stress in their nervous system [22]. GPx3 importance in stem cells is also increasing. In human blood stem cells, Gpx3 is involved in cell viability and maintains characteristics of both normal and leukemic stem cells [53]. The rigorous regulation of intracellular H_2_O_2_ by GPx3 can indirectly activate key transcription factors for cell survival via inducing kinases phosphorylation through the phosphatase inhibition [54]. Finally, GPx3 4.5-fold expression increase under hypoxic conditions indicates a crucial role of this gene in mesenchymal stem cells elevation in cell fitness [16]. Due to the regeneration capabilities of Pe, Ant and Ante, and the undeniable presence of stem cells in these tissues, it is very probable that GPx3a and GPx3b are associated with their cells’ regulation and protection.

CqGPx3s may be involved in the regulation of inflammatory response and skeletal muscle regeneration and similar processes. In injured spinal cord, the transcription factor NeuroD6 induces the expression of GPx3 and TXNL1, which effectively scavenges excessive reactive oxygen species (ROS) and attenuates inflammation [13]. In continuously stressed tissue, such as adipose tissue obtained from morbidly obese women undergoing bariatric surgery, the *GPx3* expression was three-fold higher in fat cells compared with non-fat cells, indicating an inflammatory response [15]. Knockdown of *Gpx3* in human skeletal muscle precursor cells using siRNA induced elevation in reactive oxygen species and cell death [14]. Activation of peroxisome proliferator-activated receptor γ (PPARgamma) induces *GPx3* expression, which reduces human skeletal muscle cells’ extracellular H_2_O_2_ levels, improving insulin sensitivity by increasing insulin-stimulated glucose uptake and insulin signaling, suggesting it may be a therapeutic target for diabetes mellitus [12]. Pe is homologous to skeletal muscle and the presence of GPx3 might be associated with stem cells viability. This is very important due to the capacity of complete Pe regeneration after mutilation. As in human skeletal muscle cells [14], *CqGPx3b* levels may have important implications for the regeneration of Pe muscle. In a GPx3 similar response of iBAT MBko mice [17], it is possible that GPx3b in NC and Pe of *C. quadricarinatus* under hypoxic conditions could be related to an ATP production diminishment strategy. Multiple peroxidases regulation patterns are known. In kidney embryonic cells, GPx3 was down-regulated in TXNRD1_v1 overexpressing cells [18], indicating that related peroxidases possibly have the same function in particular tissues. On the other hand, GPx3 and TXNL1 expression are induced by the transcription factor NeuroD6 in spinal cord [13] indicating coordinated expression. Likewise, GPx3, unlike other isoforms, is expressed mainly in bivalves (*Chlamys farreri*, *Patinopecten yessoensis*) during their development, acting in the antioxidant defense system for the detoxification of paralytic shellfish toxins [21]. GPx3 also increases its expression in the skin and muscle of *Labeo rohita* fish in response to infection by *Aeromonas hydrophila* and, after handling and high temperature stress in the liver of the yellow Perch (*Perca flavescens*), being a main component in the immune defense and against stressful factors [19,20].

Increased oxidoreductases during stressful conditions have been demonstrated in crustaceans [28,29]. For instance, the *L. vannamei* GPx increase, by effect of hypoxia, led to reoxygenation and high temperature stress in a conditional and specific way [55,56,57]. Specifically, changes in GSH occurred in *C. quadricarinatus* exposed to stressful conditions [27,31]. This response to transcriptional level could be enhanced by the HIF-1 transcription factor, previously identified as strong regulator of human GPx3 [58]. The upward expression of *CqGPx3a* in NC and Pe in stress events could have a similar response. The lack of response of *CqGPx3b* expression in environmentally stressful conditions; the predicted antimicrobial Pro-rich preprotein element of Pe *CqGPx3b* gene, and the significantly superior expression of *CqGPx3b* in the control group compared to *CqGPx3a*, point to a possible pathogen-specific response of *CqGPx3b*.

## 5. Conclusions

*C. quadricarinatus* possess two isoforms of *GPx3* that are expressed in a tissue-specific manner in the nervous system. *CqGPx3a* mRNA is preponderantly present in complex and protected tissues as NC, CG and Ante, while *CqGPx3b* mRNA is mainly expressed in exposed tissues as Pe. In both cases, there is a pro-peptide conformed by a cleavable signal peptide, GPx3 functional protein and a cleavable C-terminal domain. *CqGPx3a* C-terminal domain is susceptible to phosphorylation, a desirable characteristic to regulate the protein maturation process. *CqGPx3b* has a Pro-rich C-domain related to antimicrobial response. There is a lack of information about *GPxs* isoforms expression in crustaceans, and herein we report differential expression in optimal and under stress (hypoxia and temperature) conditions, where hypoxia appears to have a stronger effect in *CqGPx3a* and *CqGPx3b* expression. *CqGPx3a* isoform is more expressed than *CqGPx3b* when the crayfishes are stressed by moderate and severe temperatures as well as with hypoxia. It will be very interesting to continue the characterization of the Pro-rich C-terminal domain of *CqGPx3b* as an antimicrobial peptide. Our data indicate that both GPx3s may contribute to the maintenance of H_2_O_2_ extracellular levels associated with the regulation of a great variety of factors such as intracellular second messengers availability, insulin response, glucose accessibility, ATP synthesis strategies, stem cells differentiation, extracellular response to microbial infection and inflammatory processes. *GPx3* isoforms and their proteins are promising objectives in crustaceans to determine initial extracellular regulation of metabolic routes and cellular processes in normal as in stressful conditions.

## Figures and Tables

**Figure 1 genes-13-00179-f001:**
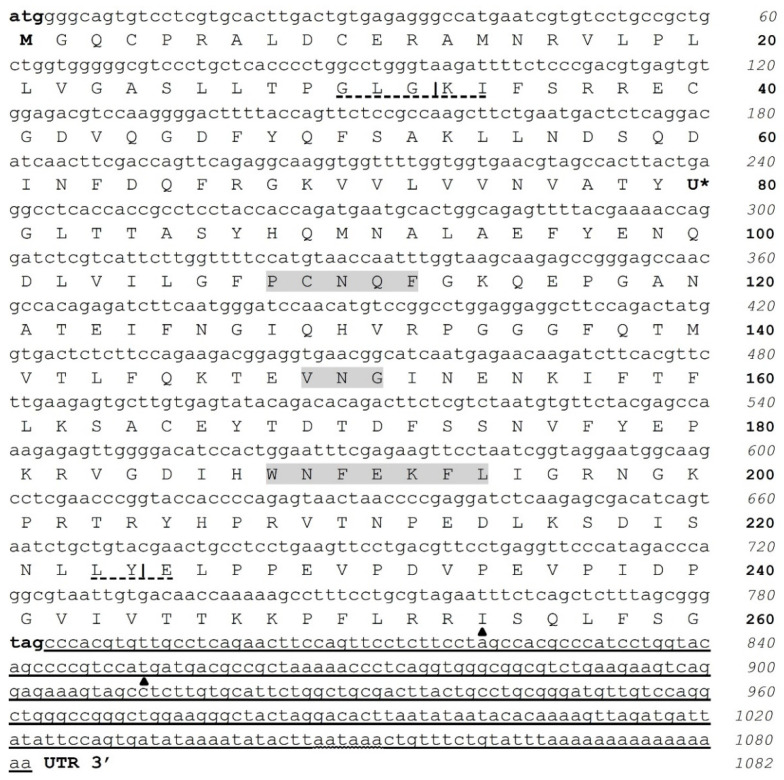
cDNA nucleotide and amino acid sequences of *C. quadricarinatus* nerve cord glutathione peroxidase 3 isoform “a” (*CqGPx3a*). Initial methionine (ATG) and stop (TAG) codons are in bold. Sec (U) is marked with *. Pro-peptide is formed by a signal peptide (32 amino acids), a mature protein (192 amino acids) and a phosphorylation site (36 amino acids). N-terminal cleavage site at position 32 (GLG-KI) and C-terminal at 224 (LY-E) are marked. GPx signature sequence motifs, PCNQF, VNG and the extra active site motif, WNFEKFL, are shown in gray; 3′ UTR is underlined. Selenocysteine Insertion Sequence (SECIS) start and end are marked by ▲ and consensus polyadenylation signal (AAUAAA) is undulated and underlined.

**Figure 2 genes-13-00179-f002:**
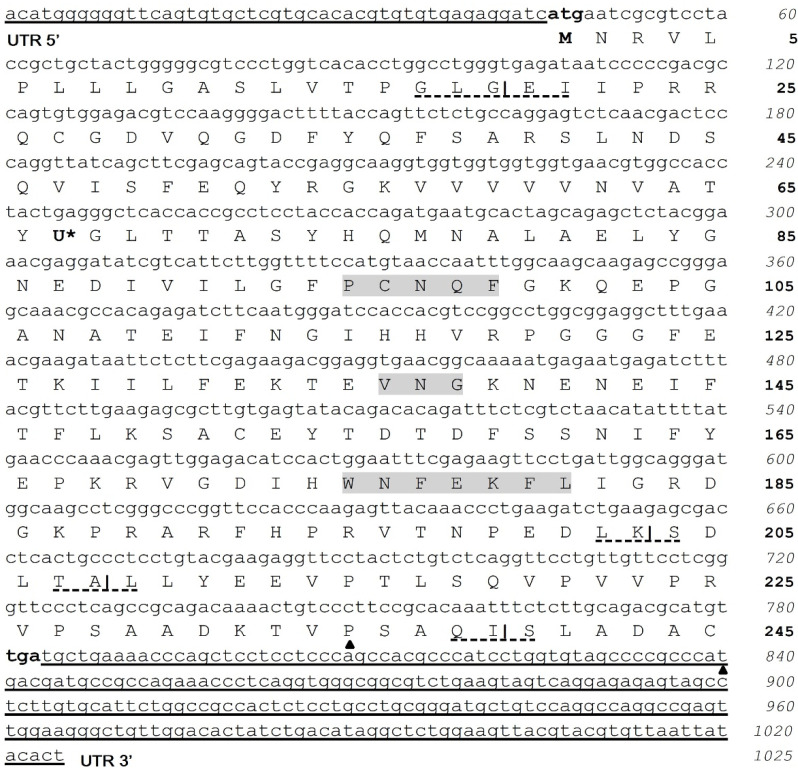
The nucleotide and amino acid sequences of *C**. quadricarinatus* pereiopod glutathione peroxidase 3 isoform “b” (*CqGPx3b*). Initial methionine (ATG) and stop (TGA) codons are in bold. Sec (U) is marked with *. Pro-peptide is formed by a signal peptide (19 amino acids), mature protein (184 amino acids) and Pro-rich peptide (42 amino acids). N-terminal cleavage site at amino acid 19 (GLG-EI) and C-terminal at amino acids 203 (LK-S), 208 (TA-L) and 239 (QI-S) are marked. GPx signature sequence motifs, PCNQF, VNG and the extra active site motif, WNFEKFL, are shown in gray; 5′ and 3′ UTRs are underlined. Selenocysteine Insertion Sequence (SECIS) start and end are marked by ▲.

**Figure 3 genes-13-00179-f003:**
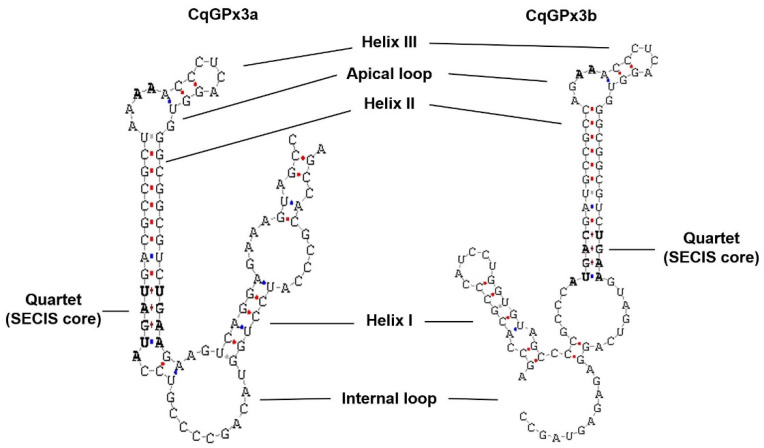
*C. quadricarinatus* glutathione peroxidase 3 isoform “a” (*CqGPx3a*) and *C. quadricarinatus* glutathione peroxidase 3 isoform “b” (*CqGPx3b*) Selenocysteine Insertion Sequence (SECIS) putative type 2 secondary structures. Conserved SECIS nucleotides are marked in bold.

**Figure 4 genes-13-00179-f004:**
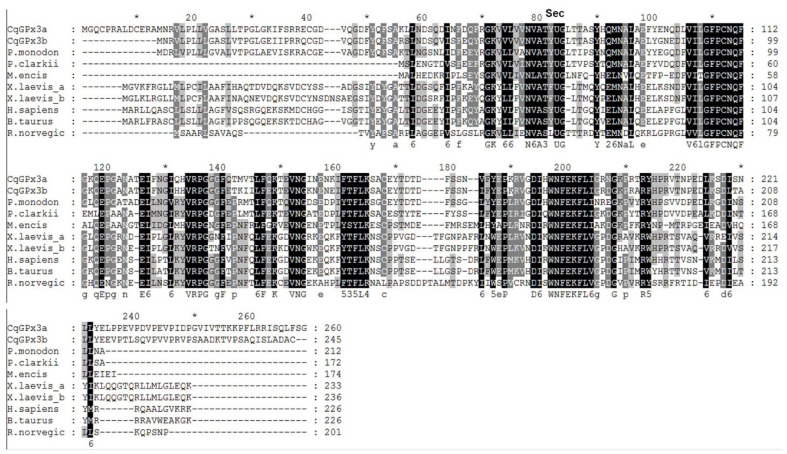
Alignment of ▲ *C. quadricarinatus* glutathione peroxidase 3 isoform “a” (*CqGPx3a*) (KX685410) and ▲ *C. quadricarinatus* glutathione peroxidase 3 isoform “b” (*CqGPx3b*) (*KX685411*) predicted proteins with other *G*lutathione peroxidases (GPxs) from *P. monodon* GPx3 (ALM09356.1), *P. clarkii* (G9JJU2.1), *Metapenaeus encis* (ACB42237.1), *Xenopus laevis* GPx3a (NP_001085319.2), *X. laevis* GPx3b (NP_001086142.2), *Homo sapiens* GPx3 (P22352.2), *B. Taurus* GPx3 (P37141.2) and *Rattus norvegicus* GPx3 (P23764.2). 100, 80, 60 and below 60 percent of identity between sequences are in black, dark gray, light gray and white, respectively. Sec indicates selenocysteine (U) residue.

**Figure 5 genes-13-00179-f005:**
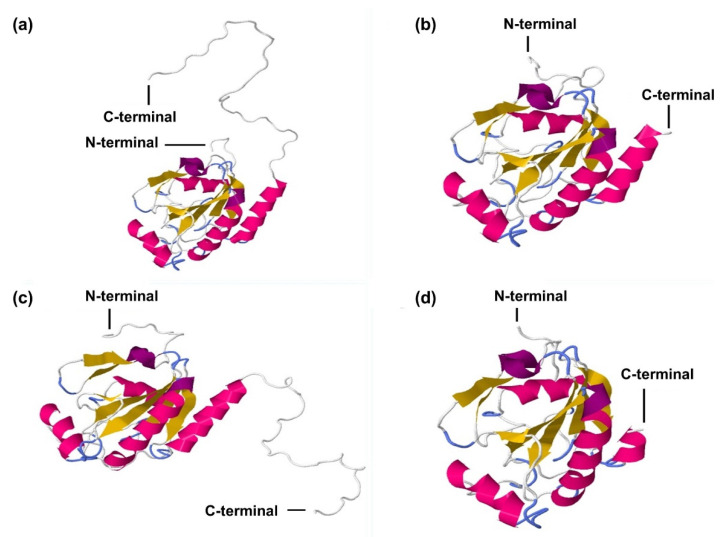
*C. quadricarinatus* glutathione peroxidase 3 isoform “a” (*CqGPx3a*) and *C. quadricarinatus* glutathione peroxidase 3 isoform “b” (*CqGPx3b*) protein three-dimensional predicted structure using RaptorX, mold PDB molecule: *Homo sapiens* glutathione peroxidase 3. PDB title: crystal structure of human glutathione peroxidase 3 (selenocysteine to glycine mutant). (**a**) *CqGPx3a* with C-Terminus, 228 residues (100% of the sequence) were modeled with *p*-value: 9.83 × 10^−11^, uGDT(GDT): 146(64) and uSeqId(SeqId): 85(37); (**b**) *CqGPx3a* without C-Terminus, 192 residues (100% of the sequence) were modeled with *p*-value: 6.47 × 10^−11^, uGDT(GDT): 145(76) and uSeqId(SeqId): 84(44); (**c**) *CqGPx3b* with C-Terminus, 226 residues (100% of the sequence) were modeled with *p*-value: 1.26 × 10^−10^, uGDT(GDT): 143(63) and uSeqId(SeqId): 83(37); (**d**) *CqGPx3b* without C-Terminus, 184 residues (100% of the sequence) were modeled with *p*-value: 6.60 × 10^−11^, uGDT(GDT): 136(74) and uSeqId(SeqId): 82(45) by the single highest-scoring template.

**Figure 6 genes-13-00179-f006:**
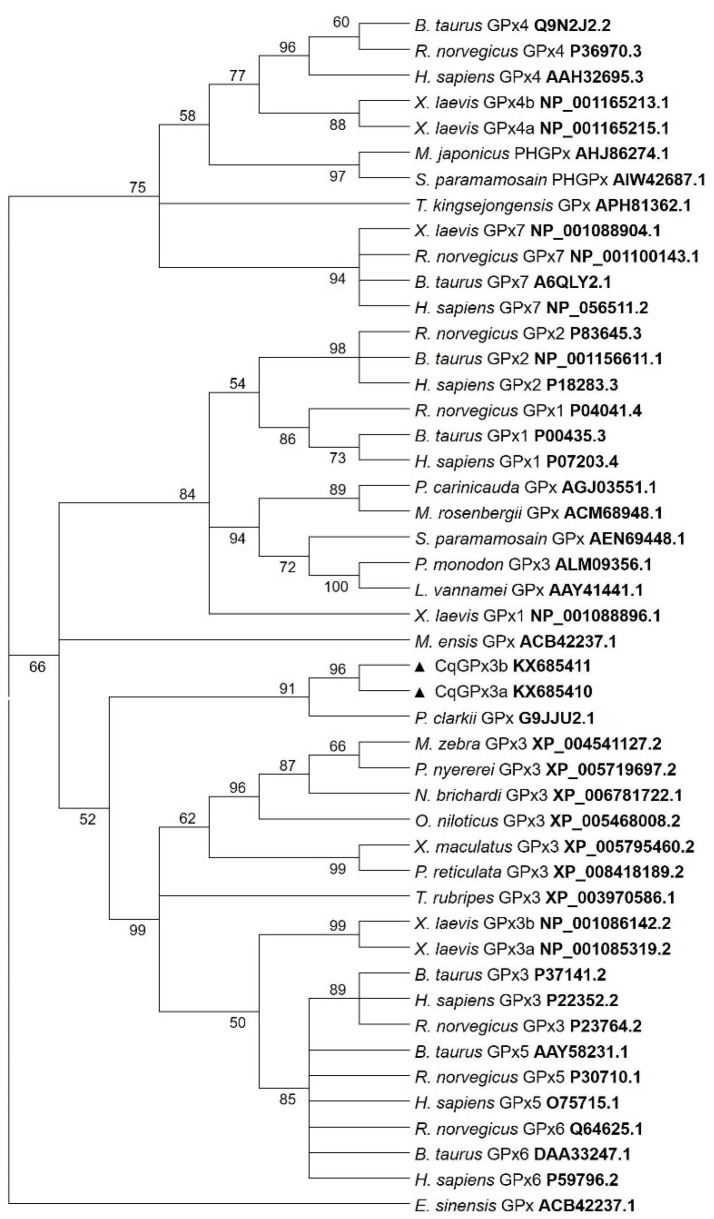
Maximum likelihood tree using the method LG with NNI as heuristic method and 500 bootstrap replicates of Glutathione peroxidases (GPxs) amino acid sequences from different species including the *C. quadricarinatus* glutathione peroxidase 3 isoform “a” (*CqGPx3a*) and *C. quadricarinatus* glutathione peroxidase 3 isoform “b”(*CqGPx3b*) protein isoforms. The branch lengths are proportional to the amino acid differences. Only bootstrap values above 50 are shown.

**Figure 7 genes-13-00179-f007:**
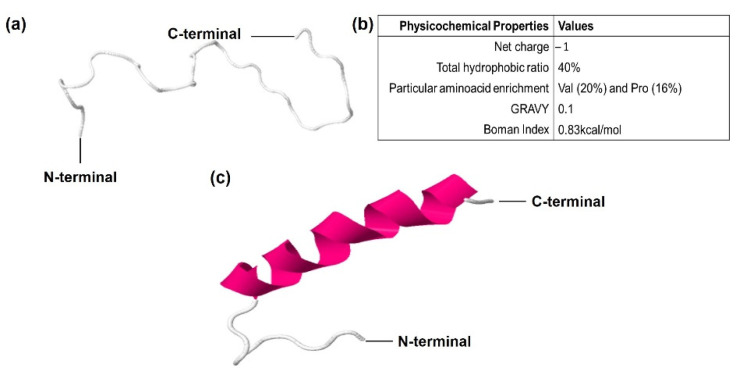
Predicted Pro-Rich antimicrobial peptide derived from *C quadricarinatus* glutathione peroxidase 3 isoform “b” (*CqGPx3b*) pre-protein isoform in peripheral nervous system. (**a**) Predicted random coiled tertiary structure, mold PDB molecule: *Drosophila melanogaster* exoribonuclease Xrn1. PDB title: crystal structure of Xrn1-substrate complex, 31 residues (100% of the sequence) were modeled with *p*-value: 1.54 × 10^−3^, uGDT(GDT): 17(55) and 100% coiled secondary structure. (**b**) Calculated physicochemical properties. (**c**) Predicted tertiary structure of Pro-rich peptide after C-terminus makes transition to α helix structure due to binding with target molecules.

**Figure 8 genes-13-00179-f008:**
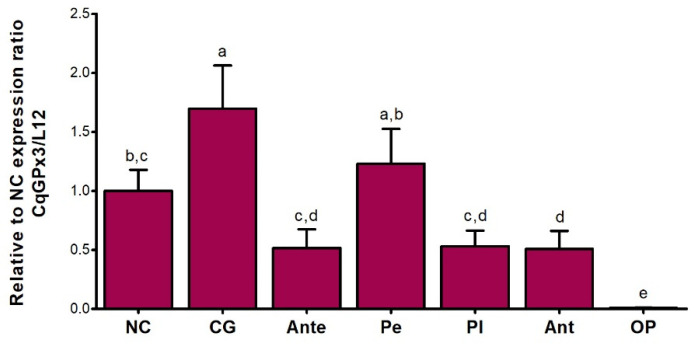
Relative expression of *C. quadricarinatus* glutathione peroxidase 3 (*CqGPx3*) relative to nerve cord in: NC: nerve cord; CG: cerebral ganglia; Ante: antennule; Pe: pereiopods; Pl: pleopods; Ant: antennae and OP: ocular peduncle by RT-qPCR. Levels of transcripts were measured by duplication. Bars represent mean ± standard deviation (*n* = 8, *N* = 56). Significant differences within tissues are indicated by letters (ANOVA, LSD’s multiple comparisons *p* < 0.05).

**Figure 9 genes-13-00179-f009:**
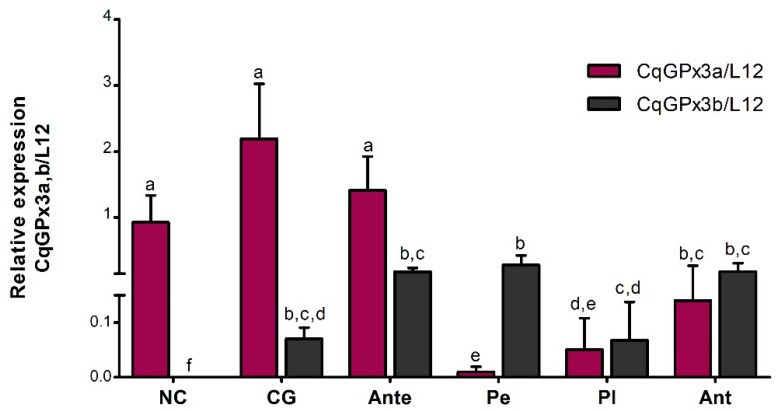
Relative quantification of *C. quadricarinatus* glutathione peroxidase 3 isoform “a” (*CqGPx3a*) and *C. quadricarinatus* glutathione peroxidase 3 isoform “b” (*CqGPx3b*) mRNA isoforms in: NC: nerve cord; CG: cerebral ganglia; Ante: antennule; Pe: pereiopods; Pl: pleopods and Ant: antennae by RT-qPCR. Levels of transcripts were measured in duplicate. Bars represent mean ± standard deviation (*n* = 8, *N* = 96). Significant differences within tissues are indicated by letters (ANOVA, Tukey’s multiple comparisons *p* < 0.05).

**Figure 10 genes-13-00179-f010:**
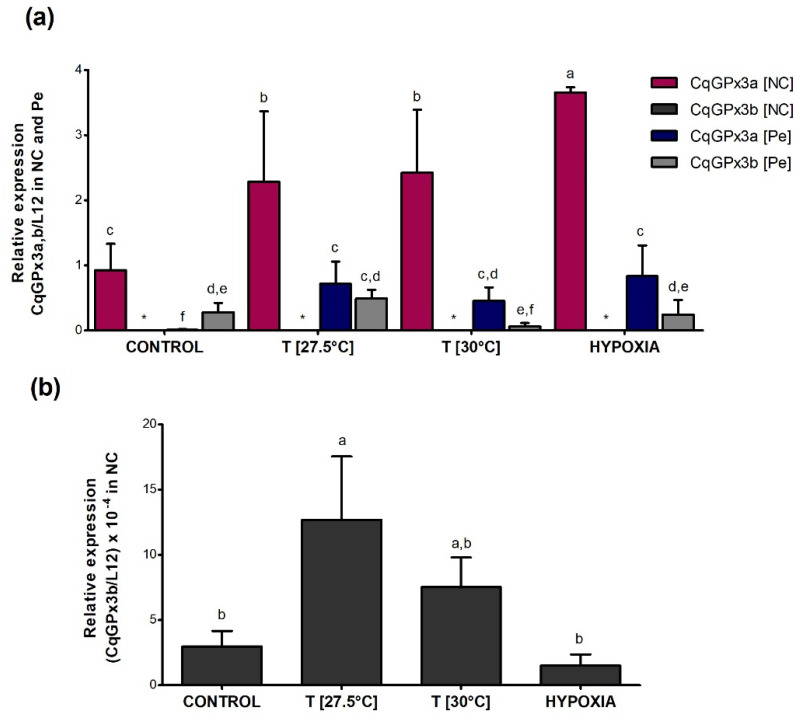
Relative expression of *C. quadricarinatus* glutathione peroxidase 3 isoform “a” (*CqGPx3a*) and *C. quadricarinatus* glutathione peroxidase 3 isoform “b” (*CqGPx3b*) isoforms in NC: nerve cord and Pe: pereiopods (**a**) and *CqGPx3b* in NC (**b**) under stress conditions by temperature of 27.5 and 30 °C and hypoxia 3.04 mg/L by RT-qPCR. Levels of transcripts were measured in duplicate. Bars represent mean ± standard deviation (*n* = 4, *N* = 15). Significant differences within treatment are indicated by letters (ANOVA, LSD’s multiple comparisons *p* < 0.05).

**Table 1 genes-13-00179-t001:** Degenerated and specific primers used.

Primer	Sequence	Amplified DNA Fragments	
GPxF2	5′-CTCGTCATTCTTGGTTTTCC-3′	*qPCR CqGPx3*	
GPxR2	5′-CCGGGTTCGGGGCTTG-3′		
GPxF4	5′-GYAAGGTRSTDYTBRT-3′	*Internal cDNA partial amplification of CqGPx3*	
GPxF5	5′-GARRRYRBVVVCAGAAWHHWC-3′		
GPxF6	5′-RGRRGGYDKWYGGBVRBRYTT-3′		
GPxR3	5′-HBYYRRGHWSKDNBYBD-3′		
GPxR4	5′-SAGCCCWRMCCMDCBBAT-3′		
GPxF7	5′- CTCGTCATTCTTGGTTTTCCATGTAACC-3′	*RACE 3′ CqGPx3a*	
GPxR5	5′- GGTTCGGGGCTTGCCATTCCTAC-3′	*RACE 5′ CqGPx3a*	
GPxF19	5′-ATATTTTATGAACCCAAACGAGTTGGA-3′	*RACE 3′ CqGPx3b*	*qPCR CqGPc3b*
GPxR12	5′-CTGAGACAGAGTAGGAACCTCTTCGT-3′	*RACE 5′ CqGPx3b*	
GPxF21	5′-GTGTTCTACGAGCCAAAGAGAGTTG-3′	*qPCR CqGPx3a*	
GPxR15	5′-CTACCCGCTAAAGAGCTGAGAAAT-3′		
L12F1	5′-CCTCTAAGTGTGTTTGCGGTGT-3′	*qPCR L12*	
L12R1	5′-AGCATCTGGTCAAGGGTCAG-3′		

## Data Availability

*CqGPx3a* GenBank accession no. KX685410, *CqGPx3b* GenBank accession no. KX685411.

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
