# Peer review of "cDNA Characterization and Expression of Selenium-Dependent CqGPx3 Isoforms in the Crayfish Cherax quadricarinatus under High Temperature and Hypoxia"

_genes, 2022, doi:10.3390/genes13020179_

Round 1
Reviewer 1 Report
It represents an important contribution to studies related to glutathione peroxidase, and its role under stressful conditions. The manuscript is well prepared and written, with the results richly presented and supported by the proposed objectives. I am convinced that the work will be useful for studies related to physiological conditions in stressful situations.
Reviewer 2 Report
The paper titled: cDNA characterization and expression of selenium-dependent CqGPx3 isoforms in the crayfish Cherax quadricarinatus under high temperature and hypoxia, by Laura E. Hernández-Aguirre et al. is very ineresting. However text of the publication is well prepared and the results are well developed (with well-developed graphic materials), the experimental part of the paper, especially the description of the research material shoud be completed. Below are some suggestions that should be included in improving the publication:
- Information about the research material requires more detail: number of individuals in the experiment / in experimental groups and in individual variants; age and sex of the crayfish, information on the date of the crayfish moult, information about crayfish maturity and health conditio (line 108).
- Characteristics of the tanks where the crayfish has been held (line 109)
- Information about the photoperiod (line 118)
- In treatement with hypoxia should be decribed with details – the parameters of oxygene should be provided in the experimental groups and controls
- If hypoxia was included in the experiment, the behavioral analysis shoud be carry out o or same behavioral observations shoud be noticed during experiment and described in the paper;
- I was wondering why the ovary did not studied.
All the parts of molecular analyses are described very well, only the experimental part about the research material should be completed.
Reviewer 3 Report
In this study, two GPx3 (CqGPx3) cDNAs (CqGPx3a and CqGPx3b) are characterized from cray fish Cherax quadricarinatus. Protein information about CqGPx3a and CqGPx3b are also characterized. CqGPx3 are expressed in cerebral ganglia, pereiopods and nerve cord. CqGPx3a is expressed mainly in cerebral ganglia, antennulae and nerve cord, while CqGPx3b was detected mainly in pereiopods. Interestingly, the expression patterns of CqGPx3a and CqGPx3b were not consistent under high temperature and hypoxia conditions. However, there are still some problems that need to be corrected in the article
Major comment
Line 204
The method of qRT-PCR data processing should be introduced in more detail. Is the relative expression calculated by the method of -2^△△CT? It's not clear here.
Line211
The tissues used by the authors here include nerve cord (NC), cerebral ganglia (CG), antennule (Ante), ocular peduncle (OP), pereiopods (Pe), pleopods (Pl) and antennae (Ant). Does nervous system tissue include pereiopods (Pe) and pleopods (Pl), or belongs to muscle?
Line217
For the convenience of the reader, the different uses of primers should be noted in Table 1, such as qPCR, PCR or RACE.
Line324
In the introduction section, the authors cite studies that indeed found that some antimicrobial peptides with antibacterial functions do have some interesting regions such as proline rich (Pro-rich) regions. However, the Pro-rich region found here has not been tested for antibacterial activities, so the antibacterial function of this region is not clear. We suggest that the author can obtain the protein of this region by artificial synthesis or in vitro expression and conduct antibacterial experiments to further clarify whether this region has antibacterial function.
Line358
Are the qPCR data in Figure 8 from these two different primers including CqGPx3a and CqGPx3b, respectively? Or is it a primer designed from a conserved region of the CqGPx3 gene?
Line374
In Figure 9, we know that CqGPx3a/L12 is marked in red, but in the PI part, CqGPx3a/L12 looks black, and Figure 9 should be adjusted to avoid misunderstanding.
Are the qPCR data in Figures 8 and 9 from the same batch of samples? If so, the two figures could actually be combined into one figure, since they are both meant to illustrate the differences in tissue expression of CqGPx3a and CqGPx3b.
In Figure 10a, the CqGPx3b gene was found to be very low expressed in nerve cord (NC) under different environmental stress conditions. Although the author obtained the expression of CqGPx3b gene in nerve cord by processing the expression data in Figure 10b, is such a low expression level meaningful?
Line468-Line479
In the Discussion section, the authors cite studies related to human GPxs to discuss the phenomenon of high expression of GPxs in the C. quadricarinatus nervous system. Perhaps it would be more appropriate to cite GPxs-related studies on aquatic animals such as teleosts here.
Line 481- Line 503
In the Discussion section, most of this paragraph is about GPx3 research in humans. Studies on GPx3 gene expression in fish and other aquatic animals under environmental stress should be cited more in the discussion.
